# High-resolution mapping of fluoroquinolones in TB rabbit lesions reveals specific distribution in immune cell types

Landry Blanc[1†], Isaac B Daudelin[1†], Brendan K Podell[2], Pei-Yu Chen[1], Matthew Zimmerman[1], Amanda J Martinot[3], Rada M Savic[4], Brendan Prideaux[1], Véronique Dartois[1*]

[1]Public Health Research Institute, New Jersey Medical School, Rutgers, The State University of New Jersey, Newark, United States; [2]Department of Microbiology, Immunology and Pathology, Colorado State University, Fort Collins, United States; [3]Center for Virology and Vaccine Research, Beth Israel Deaconess Medical Center, Harvard Medical School, Boston, United States; [4]Department of Bioengineering and Therapeutic Sciences, Schools of Pharmacy and Medicine, University of California San Francisco, San Francisco, Canada

*For correspondence:
veronique.dartois@rutgers.edu

[†]These authors contributed equally to this work

Competing interests: The authors declare that no competing interests exist.

**Abstract** Understanding the distribution patterns of antibiotics at the site of infection is paramount to selecting adequate drug regimens and developing new antibiotics. Tuberculosis (TB) lung lesions are made of various immune cell types, some of which harbor persistent forms of the pathogen, *Mycobacterium tuberculosis*. By combining high resolution MALDI MSI with histology staining and quantitative image analysis in rabbits with active TB, we have mapped the distribution of a fluoroquinolone at high resolution, and identified the immune-pathological factors driving its heterogeneous penetration within TB lesions, in relation to where bacteria reside. We find that macrophage content, distance from lesion border and extent of necrosis drive the uneven fluoroquinolone penetration. Preferential uptake in macrophages and foamy macrophages, where persistent bacilli reside, compared to other immune cells present in TB granulomas, was recapitulated in vitro using primary human cells. A nonlinear modeling approach was developed to help predict the observed drug behavior in TB lesions. This work constitutes a methodological advance for the co-localization of drugs and infectious agents at high spatial resolution in diseased tissues, which can be applied to other diseases with complex immunopathology.
DOI: https://doi.org/10.7554/eLife.41115.001

## Introduction

In human tuberculosis (TB), necrotic granulomas and cavities are the most prominent and treatment recalcitrant lesion types (*Canetti, 1955*). The major histopathological features of these lesions are a caseous or necrotic core surrounded by a cuff of immune cells including lymphocytes, epithelioid macrophages, foam cells or lipid-laden macrophages, and interspersed neutrophils and epithelial cells (*Leong et al., 2011*; *Dannenberg, 2006*). In these lesions, *Mycobacterium tuberculosis* (Mtb), the etiologic agent of TB, is found intracellularly in macrophages and foamy macrophages (*Peyron et al., 2008*), neutrophils (*Dallenga and Schaible, 2016*; *Berry et al., 2010*), epithelial cells (*Scordo et al., 2016*) and dendritic cells (*Tailleux et al., 2003*). Mtb can establish a durable infection in foamy macrophages leading to replication and/or long term survival in a dormant state (*Peyron et al., 2008*; *Russell, 2007*). TB granulomas present with increasingly abnormal vasculature

from the periphery inward (*Datta et al., 2015*). Most vessels are compressed along the lesion periphery and collapsed within the interior of the cellular rim. This leads to impaired vascular function and a gradual decrease of small molecule penetration as the distance from the granuloma outer edge increases (*Datta et al., 2015*). Vascular dysfunction culminates in a complete lack of blood vessels in the necrotic core. Mtb-infected foam cells or foamy macrophages largely concentrate along the interface between the cellular and caseous regions of the lesion (*Peyron et al., 2008*).

Efficacy and drug distribution studies in animal models of TB disease have shown that reaching adequate drug concentrations at the sites of infection is critical in achieving sterilization and clinical utility (*Irwin et al., 2014*; *Irwin et al., 2016*; *Prideaux et al., 2015a*; *Tanner et al., 2018*; *Zimmerman et al., 2017*). Using analytical approaches and MALDI mass spectrometry imaging (MSI), we previously showed that most TB drugs exhibit differential partitioning between the cellular and necrotic regions of pulmonary lesions (*Irwin et al., 2016*; *Prideaux et al., 2015a*; *Zimmerman et al., 2017*; *Prideaux et al., 2011*; *Prideaux et al., 2015b*). In these studies, TB drugs were imaged at low spatial resolution and quantified in cellular regions treated in aggregate, without taking immune cell type into consideration. Given the heterogeneous cellular composition of the cellular rim, we posit that the spatial distribution of TB drugs within the cellular compartment of lesions is a function of immune cell type, and likely reflects differential uptake in each cell type. In addition, decreased vascular function may affect drug penetration. To quantify antibiotic distribution at high spatial resolution and link the distribution patterns to immune cell types, we selected the fluoroquinolones (FQ), which constitute the mainstay of multidrug resistant (MDR) TB treatment. In MDR-TB patients, treatment success is associated with the use of FQs (*Ahuja et al., 2012*) and, not surprisingly, it follows that FQ resistance is associated with poor clinical outcome (*Bastos et al., 2014*; *Falzon et al., 2013*). Moxifloxacin (MXF), levofloxacin (LVX) and gatifloxacin (GTX), three later generation FQs, are used to treat MDR-TB, although GTX production for systemic administration was recently terminated due to its side effect profile. In addition, MXF is included in many 'universal' regimens under clinical evaluation to treat both drug susceptible and MDR-TB ((*Gillespie, 2016*; *Diacon et al., 2012*); http://www.endtb.org/clinical-trial; ClinicalTrials.gov), and thus has the potential to become a pivotal TB drug. We have recently shown that MXF kills non-replicating Mtb persisters in ex vivo caseum (*Sarathy et al., 2018*) at concentrations that are achieved clinically in caseous foci (*Prideaux et al., 2015a*; *Heinrichs et al., 2018*). Understanding the complex partitioning of MXF and other FQs in the cellular compartments of these lesions where intracellular pathogens reside constitutes one of the major remaining knowledge gaps to fully elucidate the lesion-centric pharmacokinetics and pharmacodynamics of this drug class.

Here we combine high resolution MALDI MSI with histology staining, and quantitative image analysis, to study the spatial distribution of small molecule drugs in distinct cellular clusters of TB lesions and understand spatial drug partitioning in relation to where the pathogen resides. We then recapitulate the observed distribution patterns using new drug uptake assays in human primary macrophages, foamy macrophages, lymphocytes, neutrophils, and an epithelial cell line, with the objective of developing in vitro tools to rapidly predict relative drug partitioning at the cellular level in vivo. The findings provide interesting clues on how partitioning into lung lesions can contribute to clinical efficacy, and validate immune cell uptake assays as a suitable tool to study drug distribution patterns in complex diseased tissues. The methods are not only applicable to antibiotics and host-directed therapy used in the treatment of TB and other infectious diseases, but could also be adapted to cancer and other diseases with complex immunopathology.

## Results

### FQs exhibit heterogeneous distribution in the cellular cuff of TB lesions

In previous studies, we have shown that FQs preferentially partition into the cellular layers of necrotic granulomas and cavities, compared to caseous foci, both in rabbit and human lungs (*Prideaux et al., 2015a*; *Prideaux et al., 2011*; *Prideaux et al., 2015b*). To refine the spatial distribution of this drug class in the cellular margins of granulomas, we first applied high resolution MALDI MSI. In two-dimensional MALDI MSI ion maps, MXF, LVX and GTX each formed a narrow rim of high signal intensity directly adjacent to the caseous center of the granuloma, as well as additional rings or pockets of higher intensity further outward from the necrotic center (*Figure 1A–F*). Although MALDI

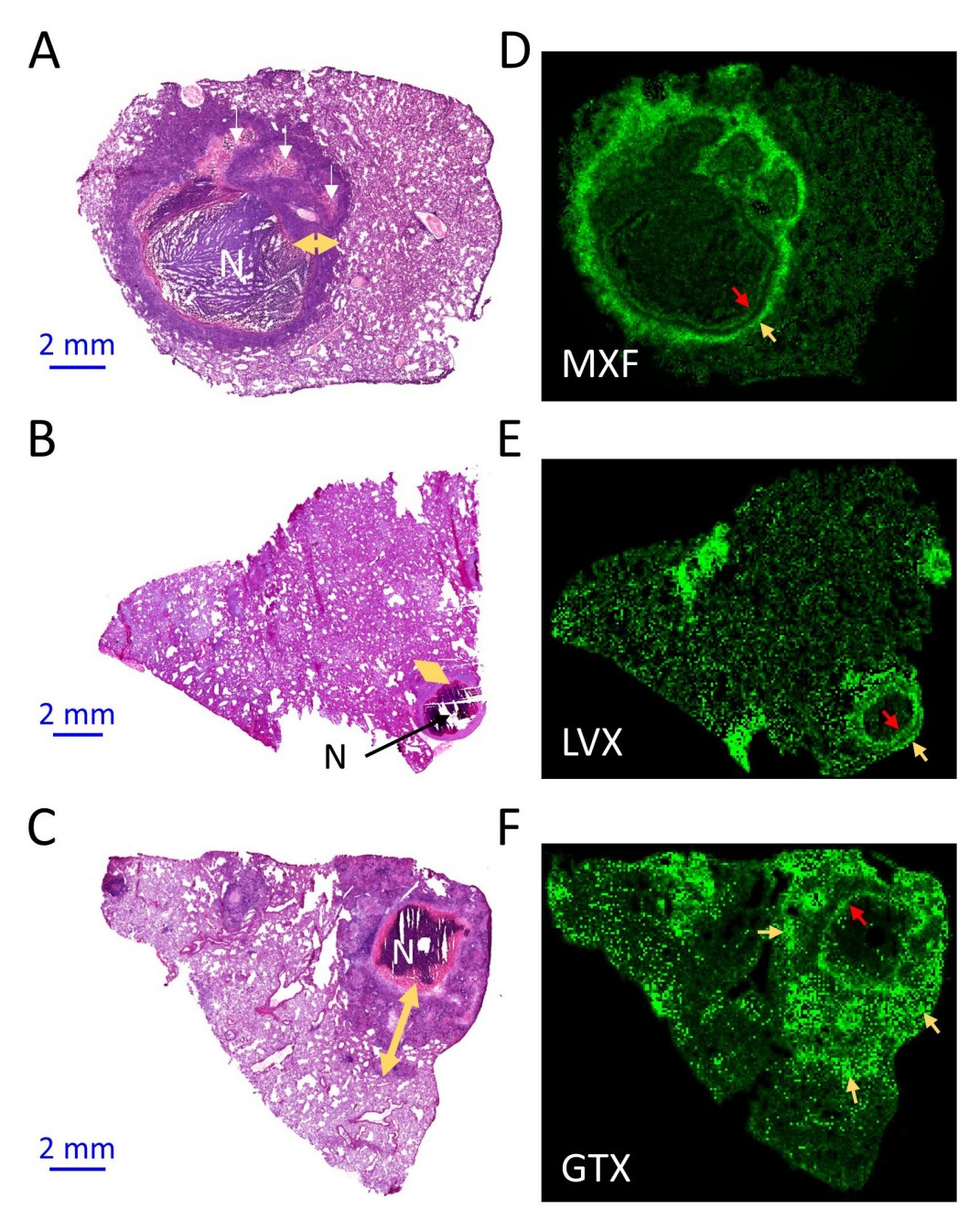

**Figure 1.** Spatial distribution of fluoroquinolones in infected rabbit lung and lesions. (a–b–c) Hematoxylin and Eosin (H and E) histology staining of lesions and surrounding lung tissue resected from rabbits that received a single dose of moxifloxacin (MXF) (a), levofloxacin (LVX) (b) or gatifloxacin (GTX) (c). N: necrotic core; white arrows: early caseating foci; yellow double arrows: cellular rim encompassing mostly lymphocytes, macrophages, foamy macrophages, interspersed epithelial cells and neutrophils. Panels d-e-f show the corresponding MALDI-MS ion maps of each drug in a tissue section adjacent to the one stained by H and E. Red arrows highlight the inner drug accumulation ring subtending the caseous core; yellow arrows highlight outer rings or pockets of higher drug abundance further outward from the core.

DOI: https://doi.org/10.7554/eLife.41115.002

The following figure supplements are available for figure 1:

**Figure supplement 1.** Schematic of the workflow for the relative quantitation of drug ions in specific areas delineated on MALDI ion maps.

*Figure 1 continued on next page*

*Figure 1 continued*

DOI: https://doi.org/10.7554/eLife.41115.003

**Figure supplement 2.** Relative quantitation of the fluoroquinolones in infected rabbit lung and lesion compartments.

DOI: https://doi.org/10.7554/eLife.41115.004

MSI does not deliver absolute drug quantitation, relative drug abundance can be compared by measuring signal intensity (drug ion counts/internal standard ion counts) in defined regions of a MALDI MSI ion map (*Figure 1—figure supplement 1*). To further characterize the visual partitioning observed for each FQ, signal intensity was measured in uninvolved lung, cellular lesion rims and necrotic foci of several lesions isolated from rabbits treated with each FQ. In this analysis, cellular and necrotic regions were treated in aggregate, and signal intensity ratios between lung, cellular and necrotic areas were calculated. We found approximately 1.5 to 2-fold higher signal in cellular lesions than in uninvolved lung and caseum for all three FQs (*Figure 1—figure supplement 2A,B*). Since this approach only delivers relative quantitation data, we next turned to laser capture microdissection coupled with standard mass spectrometry to measure the concentration of each FQ in caseous foci, cellular rims and uninvolved lung areas at various time points following an oral drug dose. Overall, MXF and GTX concentrations were higher in cellular areas than in caseum and uninvolved lung, although only marginally so. LVX exhibited irregular patterns of distribution between the cellular and caseous regions (*Figure 1—figure supplement 2C*). Thus, although MALDI images clearly reveal higher FQ abundance in distinct rings and pockets within the cellular layers of TB lesions, when the cellular rim is analyzed as a whole, the heterogeneous FQ distribution masks the favorable penetration of this drug class in the cellular cuff of mature lesions. These images and quantitative results uncovered the need for high resolution drug mapping in the cellular envelope of TB lesions.

To understand the drivers of heterogeneous FQ distribution in cellular granuloma regions, we generated high resolution H and E images of mature necrotic granulomas (*Figure 2A–F*), revealing the following concentric layers from the caseum outward: a ring of foamy macrophages with high lipid droplet contents directly subtending the caseous core, a layer mostly composed of lymphocytes, and a thick layer rich in clusters of less foamy macrophages in a background of lymphocytes (*Figure 2A–C*). Small clusters or ribbons of epithelial or stromal cells as well as interspersed neutrophils and were found in most layers (*Figure 2D–F* and *Figure 2—figure supplement 1A–F*). Mtb bacilli were mostly found in foamy macrophages and in association with karyorrhectic neutrophils (dying cells containing mostly fragmented chromatin irregularly distributed throughout the cytoplasm) in the caseous foci of necrotic granulomas and cavities (*Figure 2—figure supplement 1G,H*). When compared to the MALDI images, this organization suggested that FQs preferentially partition in histiocytes, either foamy macrophages close to the caseum or non- and less-foamy macrophages in the outer granuloma layers.

## Macrophage content and distance from lesion margin drive the penetration of MXF

Histology staining (such as H and E) of sections adjacent to those used to generate drug ion maps provide limited information as to the underlying cellular content and architecture because the two images cannot be perfectly superimposed. To overcome this limitation and correlate relative drug abundance with immune cell type, we adapted a method recently optimized by our group (*Blanc et al., 2018*), allowing MALDI MSI and H and E staining of the same tissue section. First we generated an MXF ion map and H and E stained histology image of a single section obtained from a large rabbit necrotic granuloma following treatment with a single MXF dose (*Figure 3A,B* and *Figure 3—figure supplement 1*). Using the MXF ion map, we delineated 35 regions of interest (ROI) within which drug distribution appeared relatively homogeneous (*Figure 3B* and *Figure 3—figure supplement 2A*). The MXF $[M + K]^+$ ion map was co-registered with the H and E image and the 35 ROI contour lines were redrawn onto the H and E image (*Figure 3—figure supplement 2B*). Using this image, a blinded veterinary pathologist scored the fraction of histiocytes (foamy and non-foamy macrophages), neutrophils, lymphocytes, epithelial or stromal cells, and necrosis in each ROI (*Figure 3—source data 1*). We also measured the absolute and relative distance of each ROI from the outer edge of the lesion, where the relative distance was calculated as the ratio of the distance from

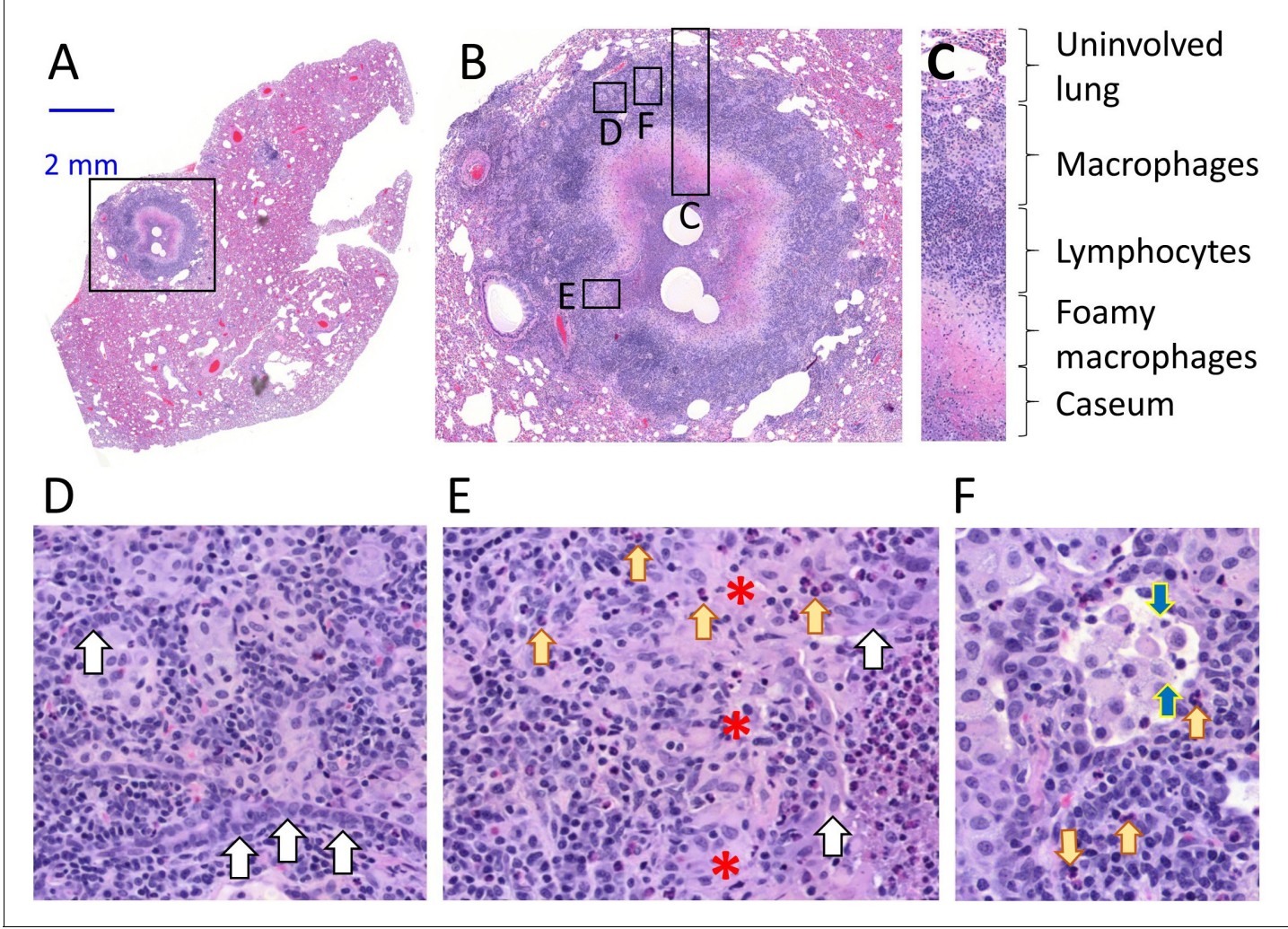

**Figure 2.** Architecture and cellular composition of typical necrotic rabbit granuloma. (A) H and E histology staining of a large cavitating necrotic granuloma and surrounding lung tissue, obtained from a rabbit infected with *M. tuberculosis* HN878 for 12 weeks. (B) Magnification of the region highlighted in a, showing the areas magnified in panels c through f. (C) Typical concentric cellular layers found in mature necrotic granulomas. (D) Magnification of the outer cellular layer showing epithelial cells that form ribbons or rings inside areas of immune cell infiltration (white arrows). (E) Magnification of the inner cellular layer showing residual epithelial lining of airway remnants (white arrows), fibrous connective tissue (red asterisks) and interspersed neutrophils (yellow arrows). (F) Magnification of the macrophage rich layer showing individual macrophage cell death (blue arrows) and isolated neutrophils (yellow arrows), surrounded by residual airway epithelium.

DOI: https://doi.org/10.7554/eLife.41115.005

The following figure supplement is available for figure 2:

**Figure supplement 1.** Architecture and cellular composition of typical necrotic rabbit granulomas.
DOI: https://doi.org/10.7554/eLife.41115.006

ROI to edge by the total distance from caseum to edge (*Figure 3—figure supplement 3*). Lastly, in the MALDI MSI ion map, we measured the relative abundance (drug/internal standard ion counts) of MXF in each ROI, expressed as the mean pixel intensity per ROI (each ROI containing 48 to 429 pixels, *Figure 3C*). We found that average MXF contents were significantly different across the majority of ROIs (*Figure 3—figure supplement 4A*) with a between-ROI variability of 43.4%. These data were compiled to search for correlations between MXF abundance, measured as mean pixel intensity, and each of the recorded parameters (i.e. fraction of histiocytes, lymphocytes, neutrophils, epithelial cells and necrotic area, and distance from lesion outer edge, *Figure 3—source data 1*), with the objective of identifying the major drivers of MXF distribution in TB lesions.

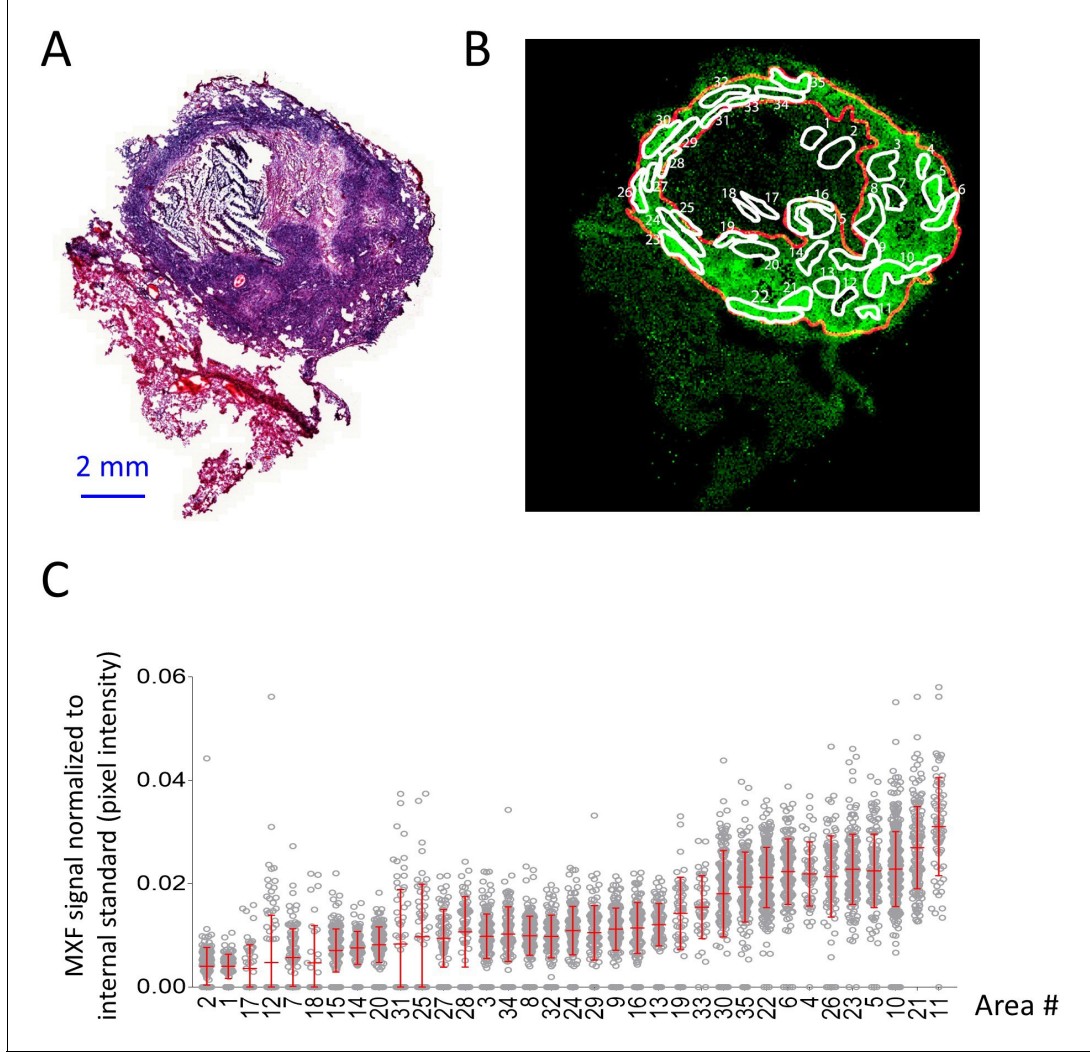

**Figure 3.** Heterogeneous distribution of MXF in the cellular cuff of a necrotic lesion. (A) H and E staining of a cavitating necrotic granuloma. (B) Ion map of moxifloxacin (MXF) [M + H]+obtained by MALDI mass spectrometry imaging of the same section, according to the workflow and procedure described in *Figure 3—figure supplement 4*; (C) relative MXF abundance in 35 sub-areas delineated in the cellular rim of the MXF ion map shown in b (Contours of the 35 sub-areas are shown in *Figure 3—figure supplement 2*). Each dot represents the signal intensity of individual pixels in the ion map shown in (B). Pixel intensity mean and standard deviation are shown for each area. The number of pixel per sub-area ranged from 48 to 429 (48 < n < 429). Raw data can be found in *Figure 3—source data 1*).

DOI: https://doi.org/10.7554/eLife.41115.007

The following source data and figure supplements are available for figure 3:

**Source data 1.** Histology and drug abundance (*Figure 3*) parameters in the 35 regions of interest, used in the covariate search and nonlinear correlation analysis (*Figure 4*).

DOI: https://doi.org/10.7554/eLife.41115.012

**Source data 2.** Spearman rank correlation, showing positive association between MXF abundance and the fraction of macrophages, and inverse correlation with relative distance from lesion border and necrosis fraction.

DOI: https://doi.org/10.7554/eLife.41115.013

**Figure supplement 1.** Workflow for overlaying MALDI-MSI and histology images.

DOI: https://doi.org/10.7554/eLife.41115.008

**Figure supplement 2.** Regions of interest in MALDI-MSI MXF ion map (A) and histology staining (B) of the same section.

DOI: https://doi.org/10.7554/eLife.41115.009

**Figure supplement 3.** Correlations between MXF abundance, distance to edge of granuloma and cell types.

DOI: https://doi.org/10.7554/eLife.41115.010

**Figure supplement 4.** Identification of the drivers of moxifloxacin distribution in cellular lesion compartments.

DOI: https://doi.org/10.7554/eLife.41115.011

Using Spearman rank correlation, we found a positive association between MXF abundance and the fraction of macrophages, and an inverse correlation with the relative distance from lesion border and the necrosis fraction (*Figure 3—source data 2* and *Figure 3—figure supplement 4B*). There was no significant association between MXF intensity signal and the percentage of other immune cell types in ROIs (*Figure 3—figure supplement 4C*). These observations prompted us to use a non-linear modeling approach to broadly interrogate the dataset and build a model of MXF partitioning in lesions. Using nonlinear mixed effect modeling (nonMEM) with the full dataset, we found that accounting for histiocyte fraction reduced the between-ROI MXF variability from 44.0% to 28.3%, which further decreased to 20.0% when relative distance of ROI to lesion edge was integrated in the model. This was consistent with vascular function and small molecule penetration decreasing as the distance from the granuloma outer edge increases (*Datta et al., 2015*). Factoring the fraction of the ROI occupied by necrotic tissue further reduced the variability from 20.0% to 18.9% (*Table 1*), significantly improving the correlation. Integrating the fraction of other immune cell types further decreased the residual unexplained variability but not significantly so (*Table 1*). This suggested that immune cells other than macrophages do not significantly influence the partitioning of MXF. The absolute distance between ROIs and lesion periphery did not correlate with MXF signal intensity, which is not surprising given the highly variable thickness of the cellular rim (*Figure 3B* and *Figure 3—figure supplement 3*). Summary plots of the covariate analysis, visual predictive checks and model parameters are shown in *Figure 4A* and *Supplementary file 1*. The outcome was consistent with the results of the Spearman rank correlation analysis (*Figure 3—figure supplement 3B,C*). The model was internally validated by randomly splitting the dataset into training and validation sets, resulting in similar model parameters (*Supplementary file 1*). The validation dataset was then used to confirm the predictive value of the model equation. We found that the model could predict ROI-specific MXF abundance using histiocyte fraction, relative distance from lesion margin and necrosis content (*Figure 4B* and *Figure 3—figure supplement 4D,E*). This suggested that MXF, and potentially other FQs, are preferentially taken up by macrophages over other immune cell types.

## In vitro uptake of FQs in immune cells is consistent with in vivo observations

In order to test this hypothesis and recapitulate these in vivo observations in vitro, we developed drug uptake assays in primary human lymphocytes, macrophages, neutrophils and in A549 human alveolar basal epithelial cells. Drug uptake in each cell type revealed that MXF, LVX and GTX all accumulated in macrophages at higher levels than in lymphocytes, neutrophils and epithelial cells in vitro (*Figure 5A*). These in vitro results were consistent with the finding that MXF relative abundance in cellular granuloma regions best correlates with the macrophage fraction. Since foamy macrophages are a privileged reservoir for Mtb bacilli in TB lung lesions (*Peyron et al., 2008*), we specifically measured FQ uptake in foamy macrophages to determine whether foamy macrophages are as FQ-avid as their non-foamy counterparts. Foamy macrophages were derived from differentiated primary macrophages via infection with γ-irradiated *M. tuberculosis*, upon which more than 80% of the macrophages appeared Nile Red positive, that is with a high lipid droplet content. In contrast, less

**Table 1.** Summary of multivariate search and model development

| | Unexplained between ROI variability in MXF abundance | P value |
|---|---|---|
| Base model – no predictors | 44.0% | |
| With relative distance from lesion border | 28.2% | $3.0*10^{-6}$ (****) |
| With fraction histiocytes (%) | 20.0% | $5.2*10^{-4}$ (****) |
| With fraction necrosis (%) (df = 3) | 18.9% | 0.05 (*) |
| With fraction lymphocytes, neutrophils, epithelial cells | 14% | ns |

df: degree of freedom; ns: not significant

DOI: https://doi.org/10.7554/eLife.41115.016

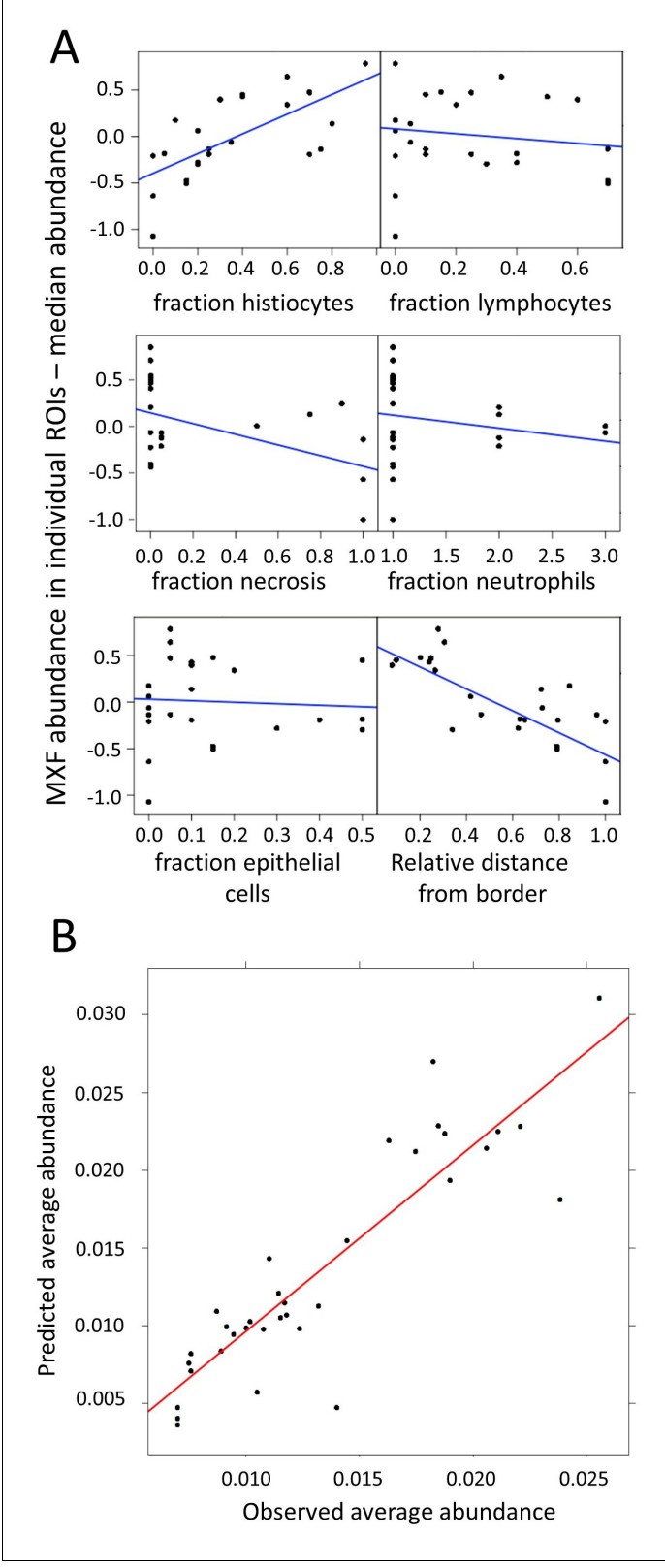

**Figure 4.** Modeling of MXF abundance as a function of histology parameters (raw data can be found in *Figure 4—source data 1*). (**A**) Diagnostic plots of Empirical Bayes Estimates (EBEs) supporting the covariate search and showing MXF abundance differences to the median (Y axes) as a function of each of the six candidate predictors, as indicated. The blue line indicates the correlation trend: positive for % histiocytes, negative for %

*Figure 4 continued on next page*

*Figure 4 continued*

necrosis and relative distance to lesion border, and neutral for the other predictors, thus illustrating the biological relationship between drug abundance and % histocyte as well as distance from the border; (**B**) Visual Predictive Checks (VPC) showing model-predicted MXF abundance versus observed abundance, indicating that drug abundance can reliably be predicted based on macrophage content, relative distance from granuloma border, and necrosis content, according the model equation:

$MXF\ abundance = \theta_1 \times e^{\theta_2 \times (Histiocytes(\%) - 0.35)} \times e^{\theta_3 \times (Distance\ Ratio - 0.65)} \times (1 + \theta_{Necrosis})$.

DOI: https://doi.org/10.7554/eLife.41115.014

The following source data is available for figure 4:

**Source data 1.** nonMEM model codes of base and final models.

DOI: https://doi.org/10.7554/eLife.41115.015

---

than 30% on the non-stimulated macrophages were Nile Red positive (*Figure 5B*). Fluoroquinolone uptake was significantly higher in foamy than in non-foamy macrophages (*Figure 5C*). Given the high inter-donor variability in absolute uptake, we also plotted the macrophage versus foamy macrophage uptake ratio for each individual donor, showing a mean ratio around 2 (*Figure 5—figure supplement 1* and *Figure 5—source data 1*). Thus FQ uptake is higher in foamy macrophages, where Mtb bacilli largely reside in vivo, than in regular macrophages in vitro.

## Discussion

In diseases with complex pathology, the distribution of drugs at the site of action is critical but challenging to study. Cellular content, vascular function and multidimensional tissue architecture all affect drug penetration in a drug class-specific manner (*Gerstner et al., 2009*; *Saggar et al., 2013*; *Rippley and Stokes, 1995*; *Dartois, 2014*; *Liu et al., 2013*). There is a need for in vivo, in vitro and in silico methods to study drug partitioning at diseased sites. In previous studies of antibiotic penetration in TB lesions, we showed higher accumulation of MXF in the cellular layers than in necrotic foci (caseum). MALDI MSI and traditional mass spectrometry analysis of human lesions revealed a correlation between diffusion into caseum and caseum cellularity (*Prideaux et al., 2015a*). Here we have combined high resolution MALDI MSI, qualification of cell types in H and E stained sections, quantitative image analysis and nonlinear modeling to show that (i) the distribution of FQs within the cellular cuff of mature TB lesions is uneven and (ii) the heterogeneous distribution of MXF in TB lesions is driven by the macrophage content, the distance from lesion border and the extent of necrosis. In vitro uptake assays in human blood cells were developed to confirm that the FQs are preferentially taken up by macrophages relative to other immune cell types typically found in TB lesions, and even more strongly so in foamy macrophages where Mtb establishes long term infection and dormancy. Our results are consistent with published data on the accumulation of FQs in macrophage cell lines (*Carlier et al., 1990*; *Michot et al., 2006*), constitute an advance over assays relying on immortal cell lines in which expression of transporters and efflux pumps can be significantly altered, and demonstrate that uptake is significantly lower in human neutrophils and lymphocytes.

High-resolution co-localization of drug and immune cell type was achieved by MALDI imaging and H and E staining of the same tissue section. While 'same-section' drug imaging and histology staining has been used to map the distribution of PIK-3 inhibitors in brain tumors (*Salphati et al., 2014*), the precise superimposition of drug and cell types hasn't been reported to date. In the cancer field, several studies have focused on drug mapping with reference to next-section H and E staining to achieve relative quantitation of drugs in ROIs but without qualification of cell populations (*Hinsenkamp et al., 2016*; *Marko-Varga et al., 2011*; *Giordano et al., 2016*).

The clear correlation between MXF abundance and relative distance from the outer lesion border is in keeping with previous studies showing that functionally abnormal vasculature increases as the distance from granuloma border increases, leading to correspondingly impaired small molecule distribution (*Datta et al., 2015*). In these studies, the authors showed that therapies thought to achieve vessel normalization, and thus improve vascular function, appear to improve small molecule delivery to TB lesions (*Datta et al., 2015*) or potentiate TB therapy (*Xu et al., 2018*). These observations are reminiscent of oncology drug penetration concepts, where excessive angiogenesis generates a

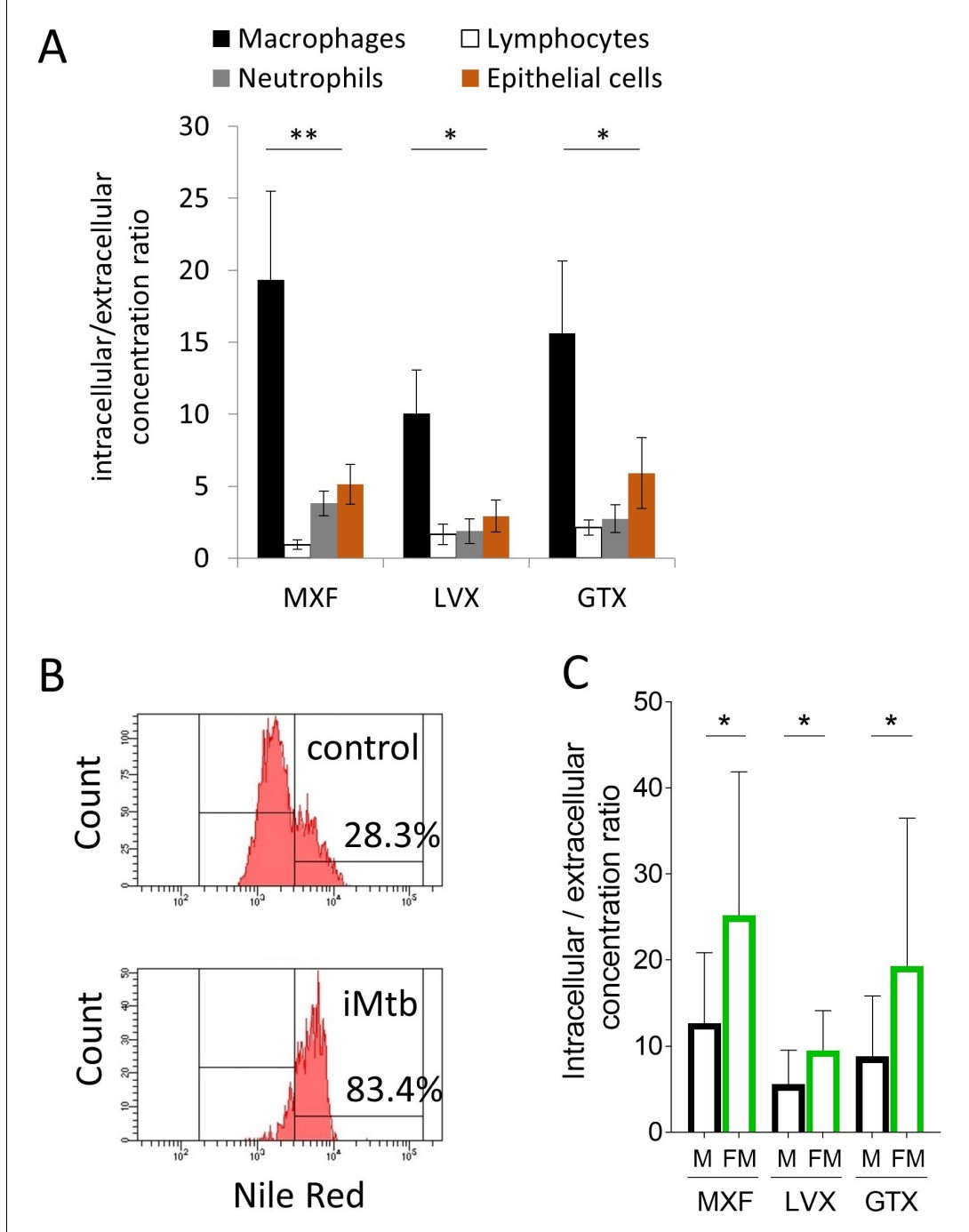

**Figure 5.** Comparative uptake of fluoroquinolones into human blood derived lymphocytes, neutrophils, macrophages and foamy macrophages, and into A549 epithelial cells. (**A**) Intracellular to extracellular concentration ratios of MXF, LVX and GTX in the major cell types present in the cellular rim of necrotic lesions. Data were analyzed using the Friedman test (all means are significantly different from each other): *p<0.05, **p<0.01; (**B**) FACS analysis of Nile Red stained human bone marrow derived macrophages showing higher frequency of stained cells in macrophage populations stimulated with heat-inactivated *M. tuberculosis* (iMtb) compared to unstimulated macrophages. The percentage of Nile Red high cells is indicated. (**C**) Intracellular/extracellular drug concentration ratio of MXF, LVX, and GTX in unstimulated bone marrow derived macrophages (black bars, (**M**), and in iMTB stimulated foamy macrophages (green bars, FM), derived from seven individual donors (raw data in *Figure 5—source data 1*). Data were analyzed using the Wilcoxon matched-pairs signed rank test. *p<0.05, **p<0.01, ***p<0.001, ****p<0.0001.

DOI: https://doi.org/10.7554/eLife.41115.017

The following source data and figure supplements are available for figure 5:

*Figure 5 continued on next page*

*Figure 5 continued*

**Source data 1.** Raw data of fluoroquinolone accumulation in macrophages and foamy macrophages obtained from the blood of 7 individual human donors.
DOI: https://doi.org/10.7554/eLife.41115.020

**Figure supplement 1.** Ratio of fluoroquinolone uptake in foamy macrophages relative to non-foamy macrophages isolated from seven individual blood donors.
DOI: https://doi.org/10.7554/eLife.41115.018

**Figure supplement 2.** Workflow for isolation and purification of lymphocytes and monocytes from donated packed leukocytes.
DOI: https://doi.org/10.7554/eLife.41115.019

disorganized and leaky blood vessel network leading to poor drug delivery to the tumor core (*Azzi et al., 2013*; *Di Paolo and Bocci, 2007*).

Interestingly, FQ uptake was higher in foamy than non-foamy macrophages. This may be due to differential regulation of transporters and efflux pumps since FQs are actively transported in and out of eukaryotic cells (*Maeda et al., 2007*; *Weiner et al., 2018*; *Mulgaonkar et al., 2013*; *Rudin et al., 1992*; *Michot et al., 2005*), and foamy macrophages are known to differentially express selected transporters such as ABCG1 and ABCA1 (cholesterol efflux transporters) compared to non-foamy phagocytes (*Lorkowski et al., 2001*; *Canfrán-Duque et al., 2017*). Alternatively, processes such as pinocytosis have been implicated in foam cell biogenesis and could be upregulated in these cells (*Michael et al., 2013*). Lastly, many FQs carry a basic amine and could undergo enhanced trapping in the acidic organelles of foamy macrophages if the pH is more acidic than in non-foamy macrophages, which remains to be determined. Regardless of the underlying mechanism, the finding that FQs are preferentially taken up by all phagocytes has positive clinical implications since drug tolerant *M. tuberculosis* persisters are present in the foam cells of human lung lesions (*Peyron et al., 2008*). On the other hand, *M. tuberculosis* bacilli are also found in viable and necrotic neutrophils, which constitute a privileged site of bacterial replication (*Mishra et al., 2017*; *Kimmey et al., 2015*; *Diedrich et al., 2016*) and where MXF uptake was significantly lower than in macrophages.

There are a few limitations to this study. If MXF uptake is higher in foamy than in non-foamy macrophages in vivo, one should detect a ring of higher signal intensity in MALDI ion maps, corresponding to the ring of foam cells that directly subtend the caseum, visible in superimposed H and E images. However, because foam cells are mostly concentrated at the caseum border where vascularization is least effective, this confounding factor prevents the visualization of higher MXF signal at the cellular-caseum interface in MXF ion maps. Second, a single lesion was used to establish the correlation between MXF abundance and histological parameters, and cell types were identified and quantified based on H and E staining which is less accurate than immunohistochemistry or flow cytometry. In addition, the method is resource intensive and not amenable to serial analyses of H and E images. This however creates an opportunity as much as it is a limitation. Our objective is to demonstrate proof-of-concept in order to harness the power of emerging technologies that rely on multiplexed ion beam imaging (MIBI-TOF) for revealing immune environment structure and composition (*Keren et al., 2018*) coupled to artificial intelligence and convolutional neural networks (*Wainberg et al., 2018*) for image recognition applied to H and E and MALDI MS images. Future efforts will focus on exploring these approaches to automate interpretation of overlaid MALDI MS, MIBI-TOF and H and E images. While the model needs to be tested with additional lesions and other FQs, the successful train-and-test approach validates the modeling strategy as a starting point to initiate similar types of investigations with other drug classes and disease indications.

The methodology is broadly applicable to the high-resolution mapping of any drug in diseased tissue, particularly when multiple cell types are present in complex lesions, tumors, atherosclerotic plaques, to name a few. The co-localization of drugs and immune cell populations supported by in vitro uptake data is a powerful new tool to guide in silico modeling and simulations of drug distribution in TB lesions (*Pienaar et al., 2017*) or any complex structure where drug penetration is paramount.

# Materials and methods

## Key resources table

| Reagent type (species) or resource | Designation | Source or reference | Identifiers | Additional information |
|---|---|---|---|---|
| Cell line (Human) | A549 | Sigma Aldrich | Catalog # 86012804-1VL | |
| Commercial assay or kit | EasySep Human CD14 Positive Selection kit | Public: StemCell Technologies | Catalog # 17858 | |

## Ethics Statement

All animal studies were performed in Biosafety Level three facilities and approved by the Institutional Animal Care and Use Committee (IACUC protocol number 16016) of the New Jersey Medical School, Rutgers University, Newark, NJ, under the guidelines and regulations of the National Institutes of Health.

Immune cells were separated and purified from fresh packed leukocytes purchased from the New York Blood Center. All samples were anonymized.

## Rabbit infection and drug administration

Female New Zealand White (NZW) rabbits (Millbrook Farm, Concord, MA), weighing 2.2 to 2.6 kg, were maintained under specific pathogen-free conditions and fed water and chow ad libitum. The rabbits were infected with *M. tuberculosis* HN878, using a nose-only aerosol exposure system as described (*Subbian et al., 2011*). Three hours post-infection, one rabbit from each round of infection was sacrificed to determine the bacterial load implanted in the lungs. At defined time points from 16 to 20 weeks post-infection, rabbits received a single dose of 100 mg/kg moxifloxacin (Chemieliva Pharmaceuticals, China), 75 mg/kg levofloxacin or 100 mg/kg gatifloxacin (Chem-Impex Intl, IL), formulated in 40% sucrose and PEG400 (90:10) by oral gavage. The time points post infection were selected to ensure that mature necrotic granulomas had formed and reached a size sufficient to allow dissection and imaging of individual lesions. Blood was collected from the central ear artery of each rabbit pre-dose, and at several time points between drug administration and necropsy. Rabbits were euthanized at 2 to 12 hr post-dose, or between the time of peak plasma concentration and the end of the tissue distribution phase.

## Tissue sectioning and processing for MALDI-MSI and laser-capture microdissection (LCM)

Tissue sections of appropriate thickness were cut from γ-irradiated rabbit lung biopsies using a Microm HN505 N (Walldorf, Germany) and thaw-mounted onto either stainless steel slides for MALDI-MSI (12 µm thick), PET-Membrane FrameSlides (Leica) for LCM analyses shown in *Figure 1— figure supplement 2C* (25 µm thick as described in *Zimmerman et al., 2018*), or standard glass microscope slides for H and E staining of adjacent slides (6 µm thick) shown in *Figure 1*. For same-section MALDI-MSI and H and E staining, 12 µm thick sections were cut and processed as described in *Blanc et al., 2018* and *Figure 3—figure supplement 1*. Tissue sections were immediately transferred to sealed containers and stored at −80°C.

Plates containing tissue sections for MALDI-MSI were allowed to reach room temperature for 15 min prior to opening of the containers. 2,5-Dihydroxybenzoic acid matrix (25 mg/mL in 50% Methanol/0.1% TFA) (Sigma-Aldrich, St Louis, MO) was applied to the tissues via the TM-Sprayer automated MALDI tissue prep device using the following optimized conditions: 0.04 mL/min flow rate; 60°C nozzle temperature; 1.3 mm/second raster speed; 25 passes over the tissue. Linezolid-d3 (TRC, Toronto, Ontario), Gatifloxacin-d4 (TRC, Toronto, Ontario) or Moxifloxacin-d4 (Clearsynth, Ontario) was added to the matrix at 5 pmol/µL as an internal standard for LZD, GTX and MXF respectively.

## MALDI-MSI analysis

MALDI-MSI acquisition was performed MALDI LTQ Orbitrap XL mass spectrometer (Thermo Fisher Scientific, Bremen, Germany) with a resolution of 60,000 at *m/z* 400, full width at half maximum. Imaging data was acquired in full scan mode to maximize sensitivity and drug peak identities were

confirmed by acquiring several MS/MS spectra directly from the dosed tissues. Instrument parameters were tuned and optimized using spiked fluoroquinolone drug standards on stainless steel plates and control mouse lung tissue. Limits of detection (LOD) were assessed as described previously (*Prideaux et al., 2015a*). The lower limits of detection were 100 fmol or 750 ng/g for LVX, GTX and MXF.

For LVX, GTX and MXF analysis, spectra were acquired in positive mode across the mass range *m/z* 300–500. A laser energy of 7.5 μJ was applied and five laser shots were fired at each position (total of 1 microscan per position). The laser step size was set at 50 μm which enabled small necrotic areas within lesions to be resolved without overlapping of the laser spot on adjacent acquisitions. Thermo ImageQuest software (v1.01) was used to reconstruct 2D ion images. Normalized ion images of LVX were generated by dividing LVX $[M + H]^+$ signal (*m/z* 362.151 ± 0.003) by LVX-d4 $[M + H]^+$ signal (*m/z* 365.168 ± 0.003). Normalized ion images of GTX were generated by dividing GTX $[M + H]^+$ signal (*m/z* 376.167 ± 0.003) by GTX-d4 $[M + H]^+$ signal (*m/z* 380.194 ± 0.003). Normalized ion images of MXF were generated by dividing MXF $[M + H]^+$ signal (*m/z* 402.182 ± 0.003) by MXF-d4 $[M + H]^+$ signal (*m/z* 406.208 ± 0.003).

## Relative quantitation of fluoroquinolones within granuloma compartments

Relative quantitation of MXF, LVX and GFX within caseum and cellular granuloma areas was performed using ImaBiotech Software Quantinetix (v 1.7, Loos, France), following the workflow shown in *Figure 1—figure supplement 1*. Areas of interest for each tissue type (lung, cellular rim, caseum) were delineated by first aligning and superimposing the MS image over the optical scan of the tissue (acquired prior to MALDI matrix deposition). The MS image layer was made transparent and the areas were drawn based upon the optical scan (*Figure 1—figure supplement 1D*) and by referral to an adjacent H and E-stained tissue section (*Figure 1—figure supplement 1C*) as a guide rather than the MALDI ion map to avoid bias in region selection.

## High-resolution relative quantitation of MXF ions in 35 regions of interest (ROI)

Thermo ImageQuest software (v1.01) was used to export MXF pixel intensity datasets into '.imzML' format. The dataset was loaded in SCiLS Lab MVS (version 2018a Core, Bruker) and normalized to MXF-d4 $[M + K]^+$ signal (m/z 444.1638 ± 0.003). A map of normalized MXF $[M + K]^+$ signal (m/z 440.1380 ± 0.002) was generated and 35 regions of interest (ROIs) were drawn with 'polygonal region' tool, with the objective of covering the full range of relative signal intensity (*Figure 3—figure supplement 2A*). The normalized MXF signal intensities of every pixel within each ROI were exported from SCiLS. Signal intensities of these different pixels are represented in *Figure 3C*. SCiLS' MXF image with area of interest was saved and loaded into Photoshop and aligned with H and E as described below, using the ion map of m/z 362.217 as the reference for alignment. After alignment, ROIs were re-drawn on the H and E image in Photoshop (*Figure 3—figure supplement 1B*). The cellular composition of each delineated ROI was assessed by a board certified pathologist, by reference to the H and E image only.

## Co-registration of MALDI MS images and H and E histology images

Co-registration of MALDI MS images and histology was essentially carried out as previously described (*Blanc et al., 2018*). Following acquisition of the MALDI MS image, the DHB (2,5-Dihydroxybenzoic acid) matrix was washed from the tissue surface by immersing the slide for 10 s in methanol/water 1:1. The tissue was then fixed by immersing for 1 hr in 4% of paraformaldehyde (in phosphate buffer saline) prior to H and E staining. The stained tissue section was scanned using a Panoramic Desk slide scanner (3D Histech) and the full resolution image (*Figure 3—figure supplement 1A*) was loaded into Adobe Photoshop CS6 (Adobe Systems). To align the H and E and MALDI images, the ion map of *m/z* 362.217, a matrix related peak, was used. This matrix-related ion is highly abundant outside of the tissue and reveals contours of the tissue border and hollow bronchioles within (*Figure 3—figure supplement 1B*). This image was loaded into Adobe Photoshop CS6. The green square in *Figure 3—figure supplement 1A* was used as an anchor to scale and overlay the MSI and H and E images. This imprint image was rescaled and aligned using tissue

contours (*Figure 3—figure supplement 1C*). Next, the green scale map of normalized MXF $[M + K]^+$ at *m/z* 440.138 (normalized to MXF-d4 $[M + K]^+$ signal m/z 444.1638 ± 0.003) was generated in Thermo ImageQuest and loaded into Adobe Photoshop CS6 (*Figure 3—figure supplement 1D*). The normalized MXF image was aligned with the matrix marker image (*m/z* 362.217) using the Thermo ImageQuest image ruler (*Figure 3—figure supplement 1E*). At this point, the MXF intensity map is aligned and overlaid with the H and E image of the same section (*Figure 3—figure supplement 1F*).

## Immune cell acquisition and purification

Fresh packed leukocytes were obtained from the New York Blood Center. The separation, purification and differentiation of major white blood cell types is depicted in *Figure 5—figure supplement 2*. Human monocytes were isolated from the blood of three independent donors by Ficoll separation and isolation of peripheral blood monocytic cells (PBMCs) followed by purification of the PBMCs with $CD14^+$ MicroBeads (StemCellTechnologies catalog #18058). Approximately $5 \times 10^5$ cells were plated in 0.5 mL of media (RPMI supplemented with 10% FBS; 1% penicillin/streptomycin; 10 ng/mL M-CSF) on 24-well plates. Cells were incubated at 37°C with 5% $CO_2$ for 2 days, after which the culture media was replaced with fresh media. After 4 more days of incubation, the culture media was replaced with either media containing either 200 µM oleic acid or a 1:5000 dilution of γ-irradiated *M. tuberculosis* (BEI Resources cat. NR-14819) or fresh media in the control wells, followed by 24 hr incubation at 37°C with 5% $CO_2$. The stock of γ-irradiated *M. tuberculosis* was vortexed with 3 mm diameter glass beads to disrupt the bacterial aggregates; the remaining large aggregates were let to sediment for 30–40 min and the supernatant was frozen at −80°C in aliquots. Since induction of foam cells by oleic acid or γ-irradiated Mtb delivered the same results in a pilot drug uptake experiment, induction with γ-irradiated was selected as more physiologically relevant.

Lymphocytes were obtained by negative selection using the $CD14^+$ MicroBeads. Cells were counted and immediately used for drug uptake.

Neutrophils were obtained using Ficoll separation on the same fresh packed leukocytes from the New York Blood Center. The fraction below the Ficoll was removed, and water lysis performed to remove the RBCs. Water lysis was performed by adding 45 mL of $diH_2O$ to the cells, solution was mixed up and down with pipette once, and very quickly 5 ml of 10x PBS were added and mixed in. Cells were pelleted at 350 G for 10 min and the process repeated up to 3 times until the pellet formed was clear (not red). Cells were counted, assessed for viability using trypan blue, and immediately used for drug uptake.

The A549 human cell line was obtained from Sigma Aldrich (86012804) and grown using Dulbecco's modified Eagles medium (DMEM) with 2% glutamine and 10% FBS in 37°C and 5% $CO_2$. Cultures were passaged every 7 days and media replaced every 3–4 days. Cultures were maintained at a density between $2 \times 10^3$ and $1 \times 10^4$ cells per $cm^2$.

## Flow cytometry

Lipid droplets in macrophages were stained with Nile Red (100 ng/ml in PBS, 1 hr incubation) and fixed in 4% PFA for 20 min on ice to prepare cells for flow cytometry. Data was acquired on a BD FACS Celesta with red blue violet laser base configuration, and analysis of data was performed using BD FACS Diva.

## Drug uptake assays in immune cells

Monocytes, lymphocytes and neutrophils were separated and purified from freshly packed human leucocytes, as described below and in *Figure 5—figure supplement 2*. Lung alveolar epithelial cells A549 were obtained from Sigma Aldrich. All cell types were cultured and differentiated when required using standard published methods.

### Macrophages and foamy macrophages

Macrophages plated on 24-well plates (50,000 cells per well) were incubated in RPMI +drug (MXF 4 µM, LVX 16 µM, GTX 4 µM or approximately 10-fold the MIC) for 30 min at 37°C, 5% $CO_2$. Macrophages were washed three times with PBS and lysed with 0.25 mL Milli-Q deionized water. Cell lysates were stored at −20°C until analyzed by LC/MS for drug quantification. To quantify the total

number of cells/well, 50 µL of each cell lysate was removed from each well and added to a clear bottom black-sided 96-well plate. 50 µL of deionized water and 100 µL of PicoGreen (Life Technologies) were added, and the plates were incubated for 2–5 min, protected from light. Fluorescence was measured at 520 nm (excitation wavelength 480 nm). Samples were blank subtracted, and cell number interpolations were made from a standard curve.

### Lymphocytes and neutrophils

Five to 20 million cells were suspended in 5 mL of RPMI cell media containing drug (MXF 4 µM, LVX 16 µM, GTX 4 µM or approximately 10-fold the MIC) in conical tubes. Cells were incubated with drug for 30 min at 37°C, 5% $CO_2$. Following incubation, cells were pelleted at 350 g for 5 min. Supernatant was poured off carefully and the pellet resuspended in 1 mL of PBS. Cell suspensions were then transferred to 1.5 mL Eppendorf tubes and pelleted at 350 g for 1 min. Supernatant was poured off carefully and pellet resuspended in 1 mL of PBS. One microliter of cell suspension was removed at this point and viable cells counted using Trypan Blue. Cells were pelleted at 350 g for 1 min, resuspended in 500 µL sterile Milli-Q deionized water for cell lysis and stored at −20°C until processed for LC/MS analysis.

### Lung alveolar epithelial cells

A549 cells obtained from Sigma Aldrich (cat # 86012804) and cultured as described above were plated on small petri dishes and incubated for 2 days at 37°C and 5%CO2 to allow cells to become well attached and become confluent. Drug was then added to the culture media to achieve desired final extracellular drug concentration (MXF 4 µM, LVX 16 µM, GTX 4 µM), and cells incubated for 30 min. Drug containing media was poured off, and cells were washed three times with PBS (PBS was added, swirled, and immediately poured off). 100 µL of 0.25% Trypsin/EDTA solution was added for 5 min to detach cells. 400 µL diH$_2$O was added, cells were scraped off, and collected in a 1.5 µL Eppendorf tube. 10 µL was immediately removed for cell counting. Cells were allowed to lyse for 1 hr in incubation, the supernatant was removed and stored at −20°C until used for LC-MS/MS analysis.

### Calculation of intracellular drug concentration

All intracellular drug uptake data are expressed as a ratio of intracellular to extracellular drug concentration (IC/EC). The starting concentration of drug added to the macrophages was used as extracellular concentration. The intracellular concentration was calculated using the drug concentration of the cell lysate determined by LC-MS/MS analysis (described below) and adjusting for the number of cells and cell volume of each particular cell type. Macrophage and foamy macrophage volume was estimated at 755.5 µm$^3$ per cell as measured in a previous study (*Chen et al., 2018*); neutrophils volume was estimated at 363 µm$^3$ per cell (*Niemiec et al., 2015*); lymphocyte volume was estimated at 173 µm$^3$ per cell (*Chapman et al., 1981*) and epithelial cell volume was estimated at 1670 µm$^3$ (*Jiang et al., 2010*). Drugs were extracted from cell lysate solutions by adding 100 µL cell lysate to 35 µL of extraction solution (32% Methanol, 68% Acetonitrile, 1 µg/ml diclofenac as internal standard) and 15 µL of 50/50 methanol/water. Extracts were stored at −80°C or analyzed immediately by LC-MS/MS

## LC-MS/MS analysis

The following analytical methods were used to quantify fluoroquinolones in plasma, tissue homogenates and laser-capture microdissected thin section areas. Neat 1 mg/mL DMSO stocks of all compounds were serially diluted in 50/50 methanol/water to create standard curves and quality control spiking solutions. Fifteen microliters of neat spiking solutions were added to 100 µL of Milli-Q deionized water, and extraction was performed by adding 35 µL of extraction solution as described above. LC/MS-MS analysis was performed on a Sciex Applied Biosystems 4000 triple-quadrupole mass spectrometer coupled to an Agilent 1260 HPLC system to quantify LVX, GTX and MXF levels in the samples. Chromatography was performed with an Agilent Zorbax SB-C8 column (2.1 × 30 mm; particle size, 3.5 µm) using a reverse phase gradient elution. All gradients used 0.1% formic acid in Milli-Q deionized water for the aqueous mobile phase and 0.1% formic acid in acetonitrile for the organic mobile phase. Multiple-reaction monitoring of parent/daughter transitions in electrospray

positive-ionization mode was used to quantify the analytes. The compounds were ionized using ESI positive mode ionization and monitored using masses MXF (402.2/358.1), LVX (362/318.5), GFX (376/261.2), and Diclofenac (296/215). Sample analysis was accepted if the concentrations of the standard and quality control samples were within 20% of the nominal concentration. Data processing was performed using Analyst software (version 1.6.2; Applied Biosystems Sciex). NZW control plasma treated with $K_2EDTA$ was obtained from Bioreclammation and used to build standard curves. LC-MS/MS analysis was performed on a Sciex Applied Biosystems 4000 triple-quadrupole mass spectrometer coupled to an Agilent 1260 HPLC system to quantify LVX, GTX and MXF levels in the samples.

## Statistical and correlation analyses

Spearman rank correlation was used to test the association between MXF abundance and each of the parameters recorded for the 35 ROIs (*Figure 3—source data 2*).

### Nonlinear Mixed Effect Modeling (nonMEM) correlation analysis

The entire dataset included 5053 observations (pixel intensities) from 35 ROIs. Each ROI contained between 48 and 429 pixels. Data points below the limit of quantitation (BLOQ) were set to 'zero'. Non-linear mixed effect modeling was applied, where two levels of random effects were implemented: a first level describing within-ROI variability and a second level of mixed effects describing between-ROI variability. All variability terms were assumed to be log normally distributed, where the median and variance terms were estimated. First, a constant baseline model was developed (the 'base' model) where the typical value of drug abundance is the median value of pixel intensity across all ROIs, and the between-ROI variability term represents the deviation of each individual ROI compared to the typical value. The covariate model was built using a step-wise model building procedure where all continuous predictors were tested using linear and non-linear functions (exponential, power, hockey stick). The predictors included into the covariate search were: % histiocytes, % lymphocytes, % necrotic cells, % neutrophils, % epithelial or stromal cells, relative distance to lesion border (or distance ratio), absolute distance to lesion border in pixels or in µm. The % necrotic cell values were categorized into four groups – no necrosis, low necrosis (<5%), medium necrosis (>5% and<90%) and high necrosis (≥90%) and treated as categorical covariates. The predictors were retained in the model if they met two criteria: (i) significance level of p<0.05 in the likelihood ratio test used to compare the goodness of fit of the model with and without the corresponding predictor, and (ii) ability to decrease unexplained between-ROI variability. To visualize the contribution of each potential predictor, we built empirical Bayes estimate (EBE) plots (*Savic and Karlsson, 2009*), based on up to 150 repeated observations in each ROI, thus representing the true measured value. The estimate of EBE shrinkage was <1%, confirming the biological relationship between drug abundance in ROIs and each of the predictors selected in the final model. Once finalized, the covariate model was validated using residual diagnostic plots and visual predictive checks. We employed model diagnostics which included plots of typical predictions and individual predictions versus observed values, as well as plots of conditional weighted residuals versus typical prediction and visual predictive checks. The base and final model codes are provided in *Figure 4—source data 1*. The final model equation describing the typical drug abundance within each ROI is shown below:

$$MXF\ abundance = \theta_1 \times e^{\theta_2 \times (Histiocytes(\%) - 0.35)} \times e^{\theta_3 \times (Distance\ Ratio - 0.65)} \times (1 + \theta_{Necrosis})$$

where $\theta_{1-3}$ and $\theta_{necrosis}$ are model estimated constants and are reported in *supplementary file 1*.

### Internal validation of the covariate model

To validate the approach, we also randomly split the dataset into a training and validation set. The data points were categorized into low, medium or high MXF abundance, to ensure that low, medium and high ranges of abundance were well represented in both sets. Within each category, data points were randomly split (2/3 and 1/3) and assigned to either the training or validation set. The training data set included 3277 measurements from 24 ROIs and the validation data set included 1776 measurements from 11 ROIs. The model was internally validated by predicting MXF abundance (both median value and within ROI variability) using ROI-specific measurements of histiocytes, distance

ratio and % necrotic cells in the final model equation (*Figure 3—figure supplement 4D,E* and *Figure 3—source data 1*)

## Statistical tests

To detect statistically significant differences in drug uptake between cell types, groups were compared using the Wilcoxon matched-pairs signed rank test (non-parametric) for single comparisons and the Friedman test (non-parametric) for multiple comparisons (GraphPad Prism). *p* values less than 0.05 were considered statistically significant. *$p < 0.05$, **$p < 0.01$, ***$p < 0.001$. The statistical analysis summarized in *Figure 1—figure supplement 2* was carried out using the Wilcoxon matched-pairs signed rank test. Data shown in *Figure 3—figure supplement 3A* were analyzed using a one-way analysis of variance (ANOVA) comparing the mean moxifloxacin abundance of each area and a Tukey post-hoc test for multiple comparisons. All data are presented as mean ±standard deviation.

# Acknowledgements

We thank M Gennaro and her team for providing guidance with flow cytometry experiments.

# Additional information

## Funding

| Funder | Grant reference number | Author |
|---|---|---|
| National Institutes of Health | U01-HL131072 | Veronique Anne Dartois |
| National Institutes of Health | R01-AI111967 | Veronique Anne Dartois |
| Bill and Melinda Gates Foundation | OPP1174780 | Veronique Anne Dartois |

The funders had no role in study design, data collection and interpretation, or the decision to submit the work for publication.

## Author contributions

Landry Blanc, Conceptualization, Data curation, Formal analysis, Validation, Visualization, Methodology, Writing—review and editing; Isaac B Daudelin, Conceptualization, Formal analysis, Validation, Visualization, Methodology, Writing—review and editing; Brendan K Podell, Visualization, Methodology, Writing—review and editing; Pei-Yu Chen, Amanda J Martinot, Methodology, Writing—review and editing; Matthew Zimmerman, Supervision, Methodology, Writing—review and editing; Rada M Savic, Conceptualization, Software, Formal analysis, Visualization, Methodology, Writing—review and editing; Brendan Prideaux, Conceptualization, Software, Formal analysis, Supervision, Visualization, Methodology, Writing—review and editing; Véronique Dartois, Conceptualization, Resources, Formal analysis, Funding acquisition, Methodology, Writing—original draft, Writing—review and editing

## Author ORCIDs

Véronique Dartois [iD] https://orcid.org/0000-0001-9470-5009

## Ethics

Animal experimentation: All animal studies were performed in Biosafety Level 3 facilities and approved by the Institutional Animal Care and Use Committee (IACUC protocol number 16016) of the New Jersey Medical School, Rutgers University, Newark, NJ, under the guidelines and regulations of the National Institutes of Health.

## Decision letter and Author response

Decision letter https://doi.org/10.7554/eLife.41115.024
Author response https://doi.org/10.7554/eLife.41115.025

## Additional files

### Supplementary files

• Supplementary file 1. (**A**) Spearman rank correlation analysis of MXF abundance in 35 regions of interest (ROIs, *Figure 3—figure supplement 2*) versus seven recorded parameters. (**B**) Final model parameters. (**C**) Model parameters obtained with training dataset only.
DOI: https://doi.org/10.7554/eLife.41115.021

• Transparent reporting form
DOI: https://doi.org/10.7554/eLife.41115.022

### Data availability

All data generated or analysed during this study are included in the manuscript and supporting files. Source data files have been provided for Figures 3, 4 and 5. Model codes are provided for the base model and full model.

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
