## [Decision Letter]

Thank you for submitting your article "High resolution mapping of fluoroquinolones in tuberculous lesions reveals immune cell type specific distribution" for consideration by *eLife*. Your article has been reviewed by four peer reviewers, one of whom is a member of our Board of Reviewing Editors, and the evaluation has been overseen by Wendy Garrett as the Senior Editor. The following individual involved in review of your submission has agreed to reveal her identity: Bree Aldridge (Reviewer #3).

The reviewers have discussed the reviews with one another and the Reviewing Editor has drafted this decision to help you prepare a revised submission.

Summary:

Effective distribution of drugs to the site of infection is critical to control TB disease, which presents with complex lung pathology, often with heterogeneous lesions at different stages of development. In their submission, Dartois and colleagues attempt to explore tissue distribution of fluoroquinolones (FQs), which are central to the treatment of drug resistant TB. Using imaging mass spectrometry, combined with histological assessment of the same sections, the authors study tissue distribution of FQs in tuberculous lesions from infected rabbits. Previous work demonstrated that FQs distribute into the cellular cuff of necrotic granulomas and in this current submission, the authors further unravel these observations. They also develop and apply interesting statistical analysis algorithms to spatially locate quantify drug distribution.

Key findings:

1) Moxifloxacin (MXF), levofloxacin (LVX) and gatifloxacin (GTX) distribute to a region adjacent to the caseous center of the granuloma.

2) FQs preferentially distribute into foamy macrophages close to the caseum or into less-foamy macrophages in the outer layers of the granuloma.

3) MXF appears to be preferentially taken up by macrophages in the granuloma cuff over other immune cell types. These observations were confirmed in drug uptake assays where FQ uptake was higher in macrophages when compared to other cell types.

4) Preferential accumulation of FQs in foamy macrophages isolated from human PBMCs.

The study is clearly presented and the following should be addressed:

Essential revisions:

1) Mention is made of "same-section" analysis by MALDI and histopathology but the workflow in Figure 1—figure supplement 1 suggests that MALDI is done on one section and histopathology is done on a distinct adjacent section. Adjacent is also mentioned in some figure legends. This becomes confusing again when the legend in Figure 3 mentions "same-section". From the text in the first paragraph of the subsection “Macrophage content and distance from lesion margin drive the penetration of MXF”, it seems clear that the same section was used for MALDI and histopathology, this should be kept consistent throughout the manuscript.

2) The authors report a 1.5 to 2-fold increase in FQ distribution in cellular lesions when compared to uninvolved lung and caseum. This difference seems relatively small. Does it translate to meaningful differences in effective killing concentrations (MICs) in the different components of the lesion? Can statistics be provided for Figure 1—figure supplement 2? What do these differences mean for bacterial clearance in these regions?

3) Better images need to be provided for Figure 2—figure supplement 1H – it's hard to see the bacilli, very fuzzy.

4) Various quantitative analyses were conducted, which are referred to and found under different sections like statistical analysis, correlation analysis and modelling. It is difficult for the reader to follow these different sections and the manuscript would benefit of a re-arrangement of Materials and methods and Results into one section called Statistical analysis where the different types of analysis could have subsections and be presented. It is accepted that some of the statistical analysis have been presented before but as an example, the modeling part is unique to this article. As such, the description of the analysis needs to be somewhat more extensive. A more detailed model development section illustrating the development of the base model and the covariate analysis as well as information about the software used and how model diagnostics were done would be good.

5) The validation step of the analysis is not clear. What does this add? The statement in the manuscript that the validation confirmed the predictive value of the model equation is somewhat questionable. A proper validation step would have required a distinct data set, generated separately from those used to build the model but with a different condition. As an example, this can include comparison of a different dose in PKPD than was used to build the model but still within the covered dose range used for model development. It would be more appropriate if all data in the current manuscript were used for model development and no validation step made. This would also most likely reduce the uncertainty in the model parameters and perhaps change the covariate analysis with respect to statistical power. Please address this.

6) It seems that immune cells in tissues were identified by their morphologic characteristics on H&E which is inaccurate. A more appropriate approach, which may be technically difficult, would be to use specific immune-stains or preferably, flow cytometric separation of cells from tissue homogenates and subsequent measurement of drug levels. Flow cytometry was performed for human blood samples but not for rabbit tissues. Given that these methods were not used, please clearly state in the Abstract and concluding remarks that only H&E staining was used to differentiate cell types in rabbits and that this is a limitation of the study.

7) The significance of the human blood data is unclear. In tissues, the drug distribution is affected by the tissue architecture, fibrosis, etc. but this is not accounted in the blood data. Also explain how this is an advance over Carlier et al. (1990) and Michot et al. (2006). How much drug is protein bound? Are these data simply reflecting drug lipophilicity? Finally, please add a statement regarding IRB approval/exemption.

8) Supplementary figures show the tissue to plasma ratios at several time-points after drug administration. However, except at early time-points (<2 hours), wouldn't these ratios be significantly affected by the plasma clearance (half-lives ~few hours)? The tissue concentration effect noted over time may simply be due to a lower plasma level (denominator) rather than a true, absolute increase in tissue levels. Can the authors address this possibility?

---

## [Author Response]

Essential revisions:1) Mention is made of "same-section" analysis by MALDI and histopathology but the workflow in Figure 1—figure supplement 1 suggests that MALDI is done on one section and histopathology is done on a distinct adjacent section. Adjacent is also mentioned in some figure legends. This becomes confusing again when the legend in Figure 3 mentions "same-section". From the text in the first paragraph of the subsection “Macrophage content and distance from lesion margin drive the penetration of MXF”, it seems clear that the same section was used for MALDI and histopathology, this should be kept consistent throughout the manuscript.

Adjacent sections for MALDI imaging and H&E were used in Figure 1, where the distribution of the three FQs is shown. From there on, same-section MALDI and H&E was carried out throughout the rest of the manuscript. We initially opted to show the ‘adjacent-section’ workflow in Figure 1—figure supplement 1, and referred to Blanc et al. (2018) for same-section workflow. However, this was not the best decision and is likely at the root of the confusion. We have now inserted a new supplementary figure describing the same-section workflow (Figure 3—figure supplement 1), reorganized and expanded the Materials and methods to better describe this workflow, and clarified the use of adjacent section versus same section throughout the manuscript.

This begs the question of why we did not use same section MALDI and H&E to achieve high-resolution mapping for all three FQs shown in Figure 1. The reason is technical and due to a significant decrease in analytical sensitivity when using glass slides for MALDI MS imaging, which is required if the same section is to be stained by H&E.

2) The authors report a 1.5 to 2-fold increase in FQ distribution in cellular lesions when compared to uninvolved lung and caseum. This difference seems relatively small. Does it translate to meaningful differences in effective killing concentrations (MICs) in the different components of the lesion? Can statistics be provided for Figure 1—figure supplement 2? What do these differences mean for bacterial clearance in these regions?

a) “This difference seems relatively small.” When cellular lesions are taken in aggregate, i.e. when FQ signal intensity is averaged across all cell types present in cellular granulomas or in the cellular cuff of necrotic granulomas, the difference in FQ abundance between cellular lesions and caseum is indeed small. The purpose of Figure 1—figure supplement 2 is to highlight that this quantitative difference is *small* in comparison to the MALDI ion maps where the visual impression suggests a more striking contrast between the high intensity pockets in the cellular rim and the caseous core. This apparent discrepancy is later on explained by the discovery that FQ uptake is much higher in macrophages than in lymphocytes, neutrophils or epithelial cells.

b) “Does it translate to meaningful differences in effective killing concentrations (MICs) in the different components of the lesion? What do these differences mean for bacterial clearance in these regions?” Since macrophages are the major cell type infected by Mtb in lung lesions, the preferential distribution of FQs in macrophage-rich pockets that are part of the cellular rim does result in higher killing efficiency of Mtb in cellular than in necrotic lesions by the FQs. This differential effect is reported in a manuscript currently in review, and provided as supporting material with the present submission (“Comparative lesion-centric pharmacology and efficacy of fluoroquinolones for tuberculosis”).

c) “Can statistics be provided for Figure 1—figure supplement 2?” These comments and questions made us rethink the way we plotted and analyzed the data. In addition to plotting concentration ratios (lung-to-cellular and caseum-to-cellular), we have also compiled the absolute values of average pixel intensities in lung, cellular and caseum ROIs, for each FQ, now provided in Figure 1—figure supplement 2B. To detect statistically significant differences between FQ signal intensities in lung versus cellular lesion, and cellular lesion versus caseum. The data were analyzed using the Wilcoxon matched-pairs signed rank test. *P* values are shown in the figure and figure legend.

3) Better images need to be provided for Figure 2—figure supplement 1H – it's hard to see the bacilli, very fuzzy.

We have replaced panel H with a clearer image at higher magnification.

4) Various quantitative analyses were conducted, which are referred to and found under different sections like statistical analysis, correlation analysis and modelling. It is difficult for the reader to follow these different sections and the manuscript would benefit of a re-arrangement of Materials and methods and Results into one section called Statistical analysis where the different types of analysis could have subsections and be presented. It is accepted that some of the statistical analysis have been presented before but as an example, the modeling part is unique to this article. As such, the description of the analysis needs to be somewhat more extensive. A more detailed model development section illustrating the development of the base model and the covariate analysis as well as information about the software used and how model diagnostics were done would be good.

We have reorganized the “Statistical and correlation analyses”section, and expanded the nonMEM modeling description, including the model codes.

5) The validation step of the analysis is not clear. What does this add? The statement in the manuscript that the validation confirmed the predictive value of the model equation is somewhat questionable. A proper validation step would have required a distinct data set, generated separately from those used to build the model but with a different condition. As an example, this can include comparison of a different dose in PKPD than was used to build the model but still within the covered dose range used for model development. It would be more appropriate if all data in the current manuscript were used for model development and no validation step made. This would also most likely reduce the uncertainty in the model parameters and perhaps change the covariate analysis with respect to statistical power. Please address this.

This is a valid point, we debated whether to present the correlation analysis as training and validation, or model building only. We agree that formal validation would require a separate dataset from another lesion. Thus we have repeated model building with the full dataset, then split the dataset in training and validation subsets, and determined that model parameters were not affected when only 2/3 of the data were used for training (compare Supplementary file 1 – Tables 2 and 3). We believe there is still merit in internally validating the model within this particular lesion, because it would allow predicting MXF signal intensity in the entire lesion based on immune cell composition and distance from granuloma edge, not only the 35 regions of interest. The training/validation approach is now shown in supplementary figures and tables only.

6) It seems that immune cells in tissues were identified by their morphologic characteristics on H&E which is inaccurate. A more appropriate approach, which may be technically difficult, would be to use specific immune-stains or preferably, flow cytometric separation of cells from tissue homogenates and subsequent measurement of drug levels. Flow cytometry was performed for human blood samples but not for rabbit tissues. Given that these methods were not used, please clearly state in the Abstract and concluding remarks that only H&E staining was used to differentiate cell types in rabbits and that this is a limitation of the study.

Indeed, the content of immune cell types in the 35 regions of interest was scored by two blinded pathologists. We agree that this method is subject to human error and is in addition slow and resource intensive. This is both a limitation and an opportunity. Our objective was to demonstrate proof-of-concept despite these limitations, uncovering the power of emerging technologies that rely on (i) multiplexed ion beam imaging (MIBI-TOF) for revealing immune environment structure and composition (Keren et al., 2018) coupled to (ii) Artificial Intelligence (AI) and convolutional neural networks (Wainberg et al., 2018) for image recognition applied to H&E and MALDI MS images. We are currently exploring these approaches in collaboration with teams at the Gates Foundation to automate interpretation of overlaid MALDI MS, MIBI-TOF and H&E images.

We have revised the Abstract to clearly state that H&E was used for cell type identification, and expanded the Discussion to highlight the associated limitations as well as the opportunity to bring the concept to the next level with MIBI-TOF and AI approaches.

7) The significance of the human blood data is unclear. In tissues, the drug distribution is affected by the tissue architecture, fibrosis, etc. but this is not accounted in the blood data. Also explain how this is an advance over Carlier et al. (1990) and Michot et al. (2006). How much drug is protein bound? Are these data simply reflecting drug lipophilicity? Finally, please add a statement regarding IRB approval/exemption.

Here we assume that by ‘blood data’, the reviewer refers to in vitro assays of FQ uptake in immune cells isolated from human blood. We acknowledge that purified human cells do not recapitulate the environment of the granuloma, such as fibrosis, tissue architecture, differential blood supply, etc. Our objective was to determine whether the preferential partitioning of FQs in macrophage-rich regions could be predicted in vitro by comparing simple drug uptake in the major immune cell types found in human blood (neutrophils, lymphocytes, monocyte-derived macrophages and foamy macrophages). We also included an epithelial cell line. If these in vitro assays would indeed predict drug behavior at the tissue level, the benefit-to-workload ratio would be high compared to in vivo studies such as those presented here.

These assays are an advance over Carlier et al. (1990) and Michot et al. (2006) in that they rely on purified leucocytes from human blood, rather than immortal cell lines in which expression of transporters and efflux pumps can be significantly altered. We have added a note in the Discussion to highlight this.

Fluoroquinolones exhibit low PPB (plasma protein binding) and low cLogP across the class (see table below, our data). While it is likely that macrophage uptake is – to some extent – a function of protein binding which in turn correlates with lipophilicity, the value of the uptake assays relies more in comparing uptake of a given fluoroquinolone in different immune cell types, than in comparing uptake of the three fluoroquinolones in a given cell type. Here we show that preferential partitioning of FQs in macrophage rich regions could be predicted in vitro by comparing drug uptake in neutrophils, lymphocytes, monocyte-derived macrophages and foamy macrophages purified from human blood.

Fresh packed leucocytes from anonymized blood donors were *purchased* (we edited the Materials and methods to indicate this important detail) from the New York Blood Center and are IRB exempt.

DrugHu PPB ^(1)^
% free (SD)Rb PPB ^(1)^
% free (SD)cLogP (ACD)MXF71.68 (0.56)69.53 (0.03)1.6LVX62.60 (0.61)61.02 (3.34)0.84GTX76.25 (2.43)78.75 (0.89)1.09

^(1)^at 2 μg/mL, or approximately the C_average_ across the dosing interval

8) Supplementary figures show the tissue to plasma ratios at several time-points after drug administration. However, except at early time-points (<2 hours), wouldn't these ratios be significantly affected by the plasma clearance (half-lives ~few hours)? The tissue concentration effect noted over time may simply be due to a lower plasma level (denominator) rather than a true, absolute increase in tissue levels. Can the authors address this possibility?

Overall, the tissue-to-plasma ratios remain quite similar throughout the dosing interval. There are slight variations but no consistent trend. This was also observed for MXF in the clinical study we published a couple of years ago (Prideaux et al., 2015). Given that FQs exhibit low protein binding, they equilibrate rapidly between the plasma and tissue compartments, but accumulate in cellular regions because they are actively transported inside cells (see the work of Paul Tulkens et al. spanning the past 20 years). The main message of Figure 1—figure supplement 2 is to show that FQ concentrations are higher in cellular than necrotic compartments, regardless of time points post dose, and regardless of plasma concentrations. We do not comment on the tissue to plasma ratios for these reasons.